# Maternal High-Fiber Diet Protects Offspring against Type 2 Diabetes

**DOI:** 10.3390/nu13010094

**Published:** 2020-12-30

**Authors:** Huishi Toh, James A. Thomson, Peng Jiang

**Affiliations:** 1Neuroscience Research Institute, University of California Santa Barbara, Santa Barbara, CA 93117, USA; 2Department of Molecular, Cellular and Developmental Biology, University of California Santa Barbara, Santa Barbara, CA 93106, USA; JThomson@morgridge.org; 3Department of Cell and Regenerative Biology, University of Wisconsin School of Medicine and Public Health, Madison, WI 53726, USA; 4Regenerative Biology Laboratory, Morgridge Institute for Research, Madison, WI 53715, USA

**Keywords:** maternal diet, fiber, protection, offspring, hiperglycemia

## Abstract

Previous studies have reported that maternal malnutrition is linked to increased risk of developing type 2 diabetes in adulthood. Although several diabetic risk factors associated with early-life environment have been identified, protective factors remain elusive. Here, we conducted a longitudinal study with 671 Nile rats whereby we examined the interplay between early-life environment (maternal diet) and later-life environment (offspring diet) using opposing diets that induce or prevent diet-induced diabetes. Specifically, we modulated the early-life environment throughout oogenesis, pregnancy, and nursing by feeding Nile rat dams a lifelong high-fiber diet to investigate whether the offspring are protected from type 2 diabetes. We found that exposure to a high-fiber maternal diet prior to weaning significantly lowered the risk of diet-induced diabetes in the offspring. Interestingly, offspring consuming a high-fiber diet after weaning did not develop diet-induced diabetes, even when exposed to a diabetogenic maternal diet. Here, we provide the first evidence that the protective effect of a high-fiber diet can be transmitted to the offspring through the maternal diet, which has important implications in diabetes prevention.

## 1. Introduction

Diabetes is one of the fastest-growing health challenges facing us today. In the last 20 years, the number of people living with diabetes increased from 151 million to over 463 million [1]; of those, 90–95% have type 2 diabetes. Type 2 diabetes is a complex disease that is poorly understood. It is characterized by insulin resistance exacerbated by beta cell dysfunction. Lifestyle intervention and current medications including insulin therapy can help regulate blood glucose levels, but in the vast majority of patients, these regimens ultimately fail. Consequently, even with treatment, diabetic patients continue to be at risk for numerous debilitating complications and have approximately a two-fold increased mortality compared to people without diabetes [2]. Hence, prevention of diabetes is a top public health priority around the world.

Diabetic risk is strongly associated with both parental history and eating habits. The odds of developing type 2 diabetes are three- to six-fold higher for adults with at least one diabetic parent compared to adults without diabetic parents [3]. However, genome-wide association studies repeatedly reveal that genetic factors insufficiently explain the observed familial aggregation [4,5], suggesting that familial risk is also modulated by non-genetic factors, which might include learned eating behaviors [6,7] as well as exposure to early-life environment [8]. The culpability of diet is well-known but not well-defined. All three classes of macronutrients—carbohydrate, fat, and protein—have been examined quantitatively and qualitatively in the context of type 2 diabetes, but consensus is sorely lacking. In general, newer studies have proposed that a low-carbohydrate diet could be more effective for managing type 2 diabetes compared to a low-fat diet [9,10]. However, fiber is an exception. Unlike simple carbohydrates, fiber is a member of the carbohydrate food group that is associated with a reduced risk of developing type 2 diabetes [11].

Low birth and one-year-old weights are strongly linked to type 2 diabetes later in life [12]. Based on this observation, Hales and Barker developed the thrifty phenotype hypothesis, which states that poor nutrition during early developmental stages may result in reduced growth of beta cells, resulting in a physiology well-adapted to poor nutrition in later life. However, a switch to improved nutrition in such individuals would lead to the development of type 2 diabetes [13]. Evidence supporting the thrifty phenotype hypothesis comes from examining the offspring of pregnant mothers during the Dutch famine (maladapted to improved nourishment) and the Siege of Leningrad (correctly adapted to malnourishment) [14]. A decade later, Bateson and Gluckman further elaborated on this phenomenon using the predictive adaptive response hypothesis, where the phenotypic plasticity during early development confers an advantage if the future environment matches the predicted environment based on early life exposures [15]. Previous experimental animal studies have similarly demonstrated adverse long-term effects using a maternal malnourished model, frequently using a low-protein (in association with higher fat or carbohydrate) diet during gestation and lactation in sheep, pigs, and rodents [16,17,18,19]. However, research to date has focused on the impact of maternal diet in increasing disease susceptibility rather than decreasing the susceptibility to type 2 diabetes.

Nile rats (*Arvicanthis niloticus*), native to tropical Africa, have a diet primarily of leaves and stems supplemented with insects, seeds, and fruits. The diabetic Nile rat model presents a similar nutritional etiology and disease progression to type 2 diabetes in humans [20,21,22,23], and thus it is an exemplary model to study how nutritional imbalances contribute to the development of type 2 diabetes. In a laboratory environment, Nile rats readily develop diabetes when fed a standard rodent chow [22], thereby referred to as a diabetogenic diet (specifically in the context of the Nile rat). Conversely, they do not develop diabetes when fed a high-fiber diet. Both diets used in our study were commercial diets based on their robustness to induce and prevent diet-induced diabetes in the Nile rat. 

We conducted a longitudinal study of 671 Nile rats to explore the consequences of two opposing maternal diets on the subsequent susceptibility of type 2 diabetes in the offspring. To avoid potential confounding effects of regularly fasting our animals in a long-term study, we used random blood glucose (RBG) measurements to assess the development and progression of diabetes. Previous studies have compared different measurements for diabetes assessment (RBG, fasting blood glucose, and oral glucose tolerance test) in the Nile rat and demonstrated that RBG is the most reliable early biomarker and a preferred clinical parameter for predicting both incidence of diabetes and its severity in the Nile rat [24].

In the lifelong susceptibility to type 2 diabetes, both exposure to maternal nutrition and self-acquired nutrition act in concert. Previous experimental studies have investigated maternal diet as an isolated variable [25,26], but failed to incorporate offspring diet as a covariate. Additionally, we have included sex as a biological variable, acknowledging the presence of sexual dimorphism in diet-induced type 2 diabetes [27,28]. Using a simple Bayesian network, we incorporated three factors—maternal diet, offspring diet, and sex—in our statistical model to examine the probability of diabetes caused by each factor. Our aims are to provide a definitive study of how maternal diet (early-life environment) juxtaposed with offspring diet (later-life environment) shifts the long-term risk susceptibility and to clarify the role of maternal diet in the prevention of type 2 diabetes.

## 2. Materials and Methods 

### 2.1. Animals

All animal experiments were approved by the University of California Santa Barbara, Institutional Animal Care and Use Committee, and conducted in accord with the NIH Guide for the Care and Use of Laboratory Animals. Our founder Nile rats were derived from the Brandeis University colony of the KC Hayes Laboratory. Nile rats in UCSB are housed at 21–26 °C in a conventional facility with individually ventilated cages and are provided autoclaved Sanichips as bedding material.

### 2.2. Ethical Statement

The Institutional Animal Care and Use Committee of the University of California Santa Barbara approved the study protocol (ID: 893; 12/03/2014) in accordance with the NIH Guide for the Care and Use of Laboratory Animals. All applicable international, national, and institutional guidelines for the care and use of animals were followed. 

### 2.3. Experimental Animal Protocol

Nile rat dams were randomly fed a high-fiber diet (Lab Diet 5L3M; Newco Specialty, Rancho Cucamonga, CA, USA) or a diabetogenic diet (Formulab Diet 5008; Newco Specialty, Rancho Cucamonga, CA, USA) [22]. The percentage of crude fiber was 23% for the high-fiber diet and only 4% for the diabetogenic diet. The ratios for the percentage of calories provided by carbohydrate, fat, and protein were 67:10:23 for the high-fiber diet and 56:17:27 for the diabetogenic diet. Littermates were randomly assigned to either the high- fiber or diabetogenic diet, which was maintained throughout the study. Using sibling-matched diet assignments allowed us to redistribute possible genetic bias into each of the high-fiber or diabetogenic diet groups. Random blood glucose (RBG) levels were measured every four weeks starting at weaning age (4 weeks old). Diabetic Nile rats were defined as having RBG >100 based on prior studies [24]. To reduce adverse events from diabetic complications, Nile rats with RBG >500 were euthanized. The number of Nile rats grouped by diet scheme and sex is provided in Appendix A, and details from each Nile rat are provided in Appendix A. In a laboratory environment, Nile rats readily develop diabetes when fed a standard rodent chow [22], thereby referred to as a diabetogenic diet (specifically in the context of the Nile rat). Conversely, they do not develop diabetes when fed a high-fiber diet. Both diets used were commercial diets based on their robustness to induce and prevent diet-induced diabetes in the Nile rat.

### 2.4. Modeling the Probability of Diabetes

We used a simple Bayesian model framework [29] consisting of three variables (maternal diet, offspring diet, and sex) to model the probability of diabetes from 4 to 40 weeks at 4-week intervals (Appendix A). The Bayesian network parameter learning and probability inference were implemented by R package (“bnlearn”) [30]. 

### 2.5. Estimating Human Equivalent Age of Rodents

To provide human relevance, we presented in parallel the human equivalent age of rodents obtained from the Jackson Laboratory (life phase equivalencies between human and mouse) [31].

## 3. Results

### 3.1. Offspring on Diabetogenic Diet were Protected against Diabetes when Mothers Consumed A High-Fiber Diet

We observed a significant difference between the diabetogenic diet offspring from mothers that consumed a high-fiber diet compared to a diabetogenic diet. As shown in Figure 1, in the group where offspring and mothers shared the same diabetogenic diets, the average RBG depicted an upward trend from 4 weeks old and clearly reflected a diabetic value from 12 weeks old. The upward trend continued until around 36 weeks old, when the average RBG plateaued at around 300 mg/dL. In contrast, in the group with mothers on a high-fiber diet, the average RBG wavered around 100 mg/dL and did not reach a diabetic value until around 28 weeks, suggesting a delay in the onset of diabetes of about 16 weeks. Furthermore, with each subsequent reading at a later timepoint, the difference between the diabetogenic maternal diet group and the high-fiber maternal diet group became larger, suggesting that, in addition to the delayed onset of hyperglycemia, the progression of diabetes was additionally slowed down.

### 3.2. Offspring on A High-Fiber Diet did not Develop Diabetes even when Mothers Consumed A Diabetogenic Diet

Interestingly, a high-fiber diet conferred long-term protection to offspring against diabetes regardless of maternal diet. In offspring that were fed a high-fiber diet after weaning, the average blood glucose trajectory between those that were from mothers with a high-fiber diet or those that were from mothers with a diabetogenic diet was similar. The average RBG in both groups never reached the diabetic threshold of RBG > 100 mg/dL, from 4 to 40 weeks old (Figure 2). This pattern was observed in both sexes (Appendix A).

### 3.3. Protective Effect of the High-Fiber Maternal Diet was more Pronounced in Males

The blood glucose lowering effect of the maternal high-fiber diet on diabetogenic diet offspring was observed in both sexes. Males benefitted more because they had a more severe hyperglycemic profile compared to the females when on a diabetogenic diet. However, the maternal high-fiber diet protection lowered the mean RBG levels to a similar normal level in both male and female offspring (Figure 3).

### 3.4. High-Fiber Maternal Diet Reduced the Long-Term Risk Susceptibility of Diet-Induced Diabetes

In our experimental design, we have three factors—maternal diet, offspring diet, and sex—that affect the risk susceptibility of diet-induced diabetes. To account for all three factors, we built a statistical model using a simple Bayesian network from 671 Nile rats with RBG values ranging from 4 to 40 weeks old. From our model, we calculated the probability of diabetes in diabetogenic diet offspring within the high-fiber maternal diet group compared to the diabetogenic maternal diet group. As shown in Figure 4, the probability of diabetes in the diabetogenic maternal diet group increased from 28.74% at 12 weeks old (equivalent to 20 years old in a human) to 81.77% at 36 weeks old (equivalent to 38 years old in a human). In comparison, the high-fiber maternal diet group increased from 13.21% at 12 weeks old to 32.87% at 36 weeks old. These data are replotted according to sex in Figure 5.

## 4. Discussion

Effective prevention measures are needed to curb the steep rising rate of type 2 diabetes. Susceptibility to type 2 diabetes is subjected to cumulative events throughout life, possibly starting from conception, described as a life-course perspective. Early-life development in utero and during infancy has been identified as a sensitive period of phenotypic plasticity where maternal nutrition can impact the risk susceptibility of type 2 diabetes in later life. Thus, a better understanding of how maternal diet modulates disease susceptibility would inform us on effective ways to prevent type 2 diabetes.

Here, we use two opposing diets that either induce or prevent diabetes in the Nile rat to understand how maternal nutrition shifts the disease risk of type 2 diabetes. Our experimental design is different in a few ways. First, we used a lifelong maternal diet in the Nile rat dams. In other experimental animal studies, the maternal diet is typically restricted to pregnancy and lactation. However, the influence of maternal diet could include the period of oogenesis that occurs during preconception. We also considered the possibility that abruptly switching diets just prior to conception may trigger altered behavioral changes due to interrupted eating habits not pertinent to maternal nutrition. Second, maternal diet and offspring diet were treated as two independent variables, providing us an opportunity to observe the interplay between early-life nutrition prior to weaning and later-life nutrition after weaning. This added layer of information cannot be obtained from studies using maternal diet as the sole experimental variable. Additionally, we considered sex as a biological variable, versus the majority of diabetic studies in animal models where only males are represented. Third, the risk susceptibility of diet-induced diabetes was analyzed in a longitudinal manner. Diet-induced diabetes has a complicated disease profile where onset and severity are highly variable, both in the human population and the Nile rat model. Thus, we generated a detailed longitudinal dataset to capture dynamic changes over time. Additionally, our dataset for this study is unusually large, consisting of RBG values from 671 Nile rats from 4 to 40 weeks old. This allows us to validate our findings using a robust Bayesian model with three nodes: maternal diet, offspring diet, and sex.

We found that a maternal high-fiber diet reduced the probability of diabetes in offspring that consumed a diabetogenic diet after weaning (Figure 4). Although the ratios of macronutrients were not dramatically different, the percentage of crude fiber was 23% for the high-fiber diet and only 4% for the diabetogenic diet. The beneficial effect of fiber on diet-induced diabetes in the Nile rat had previously been demonstrated by Bolsinger [24]. Prospective cohort studies like the Stockholm Diabetes Prevention Program study have found that consumption of whole grain protects against the development of prediabetes and type 2 diabetes [32]. Additionally, a recent epidemiological meta-analysis reported a 10–20% reduction in type 2 diabetes incidence associated with higher dietary fiber using data from 48,468 people in 17 studies [33]. The observed protective effect could be due to the presence of dietary fiber that promotes an anti-diabetic microbiome by increasing the amount of short-chain fatty acids [34]. Another possibility is that fiber reduces the intake of calories, and the high-fiber diet is essentially a low-calorie diet. Future studies using purified diets can be used to assess how dietary fiber confers protection against type 2 diabetes. Additionally, offspring from mothers on diabetogenic diets are, on average, 1.5-fold heavier than offspring from mothers on high-fiber diets at the time of weaning (Appendix A). Perhaps a maternal high-fiber diet protects offspring from diet-induced diabetes by controlling weight gain during early development. Whereas Nile rat dams on high-fiber diets never developed diet-induced diabetes, 28.3% of Nile rat dams on a diabetogenic diet were diabetic prior to pregnancy, and this may additionally impact the offspring from mothers on diabetogenic diets.

The focus of maternal diet (marker of early-life environment) stems from studies on adults that lived through famines where low birth weight was consistently associated with detrimental metabolic health consequences in adult life [35]. To explain these observations, the thrifty phenotype hypothesis and the predictive adaptive response hypothesis were proposed, where a mismatch between early-life environment and later-life environment resulted in increased susceptibility to type 2 diabetes. However, whether it is the mismatched element or the nature of the environmental factor(s) is still controversial. A recent human study investigating different combinations of early- and later-life environmental conditions revealed that adverse early-life conditions were detrimental to later health across all environments, not only in mismatched situations [36]. Our two-by-two factorial design provided two groups with mismatched diets and two groups with matched diets. We found that Nile rats with the high-fiber maternal diet and diabetogenic offspring diet had a diabetes profile that was better than Nile rats with matched diabetogenic diets but worse than Nile rats with matched high-fiber diets. Furthermore, Nile rats with the diabetogenic maternal diet and high-fiber offspring diet did not get diabetes, similar to Nile rats with matched high-fiber diets. Therefore, our findings suggest that it is not the mismatched element that leads to type 2 diabetes; rather, it depends on the nature of the environmental cue. In the Nile rat model, a high-fiber diet is beneficial to prevent diet-induced diabetes whether exposed to it through maternal feeding or consumed later in life.

It would be interesting to see if the protective effect of a maternal high-fiber diet can be observed in other models of type 2 diabetes and the human population. Given the species difference, we expect differences in the susceptibility of diet-induced diabetes, but we postulate that the overall trends in our major findings should be robust regardless of species. One limitation of this study is that we did not record the daily food consumption and calculate the fiber intake. Nile rats frequently nibble or crumble their food, and their cages are heavily littered with food waste every day, making it difficult to determine the amount of food consumed. The variations in daily fiber intake among different individual Nile rats can potentially affect the outcomes (e.g., diabetic risk). However, we expect that the overall trend should be robust given the large sample size in this study.

In conclusion, we showed evidence that by improving early-life environment, such as using a maternal high-fiber diet, protective factors are transmitted to the offspring, leading to a remarkable reduction in the risk of susceptibility to type 2 diabetes.

## Figures and Tables

**Figure 1 nutrients-13-00094-f001:**
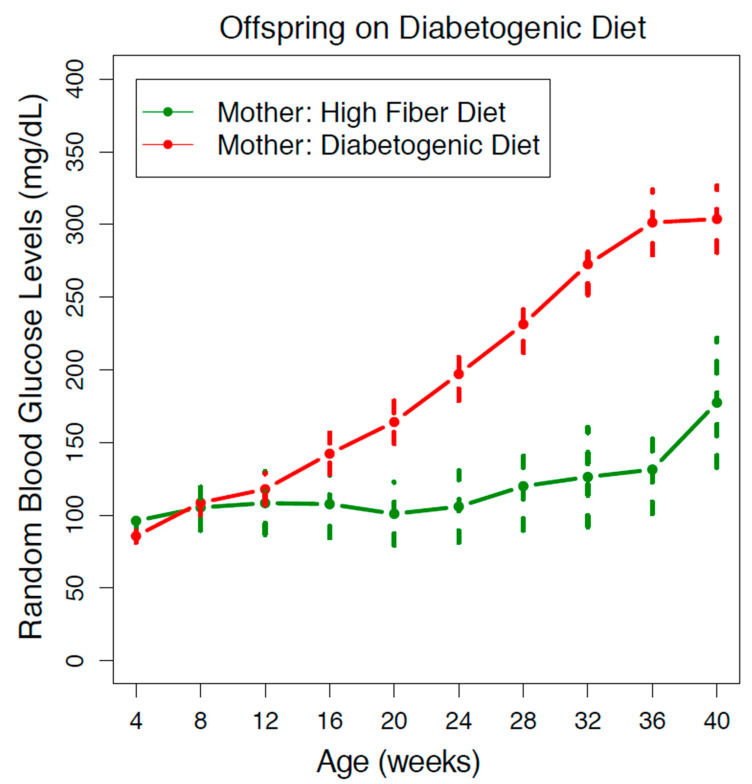
Maternal high-fiber diet protects offspring on diabetogenic diet. The progression of diabetes in the Nile rat offspring (both sexes) fed a diabetogenic diet is depicted as a series of random blood glucose levels from 4 to 40 weeks old. The offspring of mothers with a high-fiber diet, on average, experienced a delayed onset of diabetes and progressed more gradually to higher levels of RBG compared to offspring of mothers with a diabetogenic diet. The error bars reflect a 95% confidence interval of the data. The number of Nile rats: *n* = 95 (green); *n* = 335 (red).

**Figure 2 nutrients-13-00094-f002:**
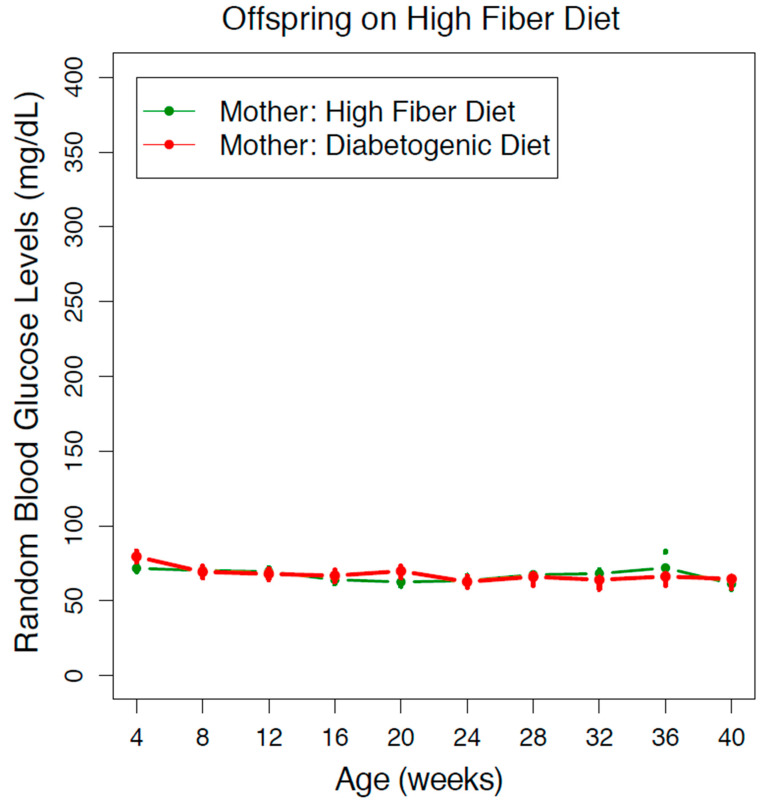
Maternal diet does not influence offspring on a high-fiber diet. The progression of diabetes in the Nile rat offspring (both sexes) fed a high-fiber diet is depicted as a series of random blood glucose levels from 4 to 40 weeks old. When the offspring were fed a high-fiber diet, they did not develop diabetes from mothers with a high-fiber diet or from mothers with a diabetogenic diet. The error bars reflect a 95% confidence interval of the data. The number of Nile rats: *n* = 131 (green), *n* = 90 (red).

**Figure 3 nutrients-13-00094-f003:**
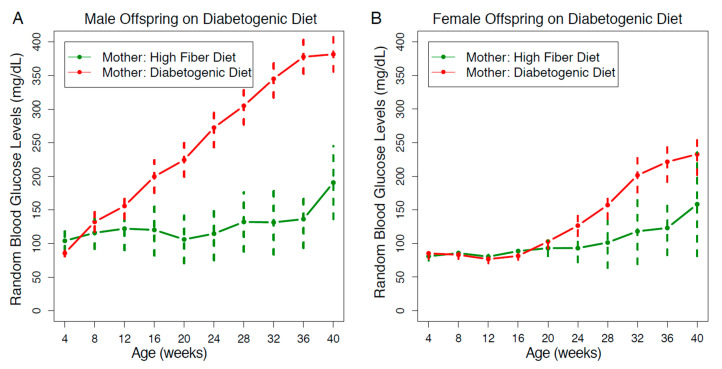
The decrease in RBG trajectory from a maternal high-fiber diet was more obvious in the male offspring. (**A**) In male offspring on a diabetogenic diet, the protective effect from a maternal high-fiber diet was evident from 12 to 40 weeks old. The number of Nile rats: *n* = 62 (green), *n* = 183 (red). (**B**) In female offspring on a diabetogenic diet, the protective effect from a maternal high-fiber diet was evident from 24 to 40 weeks old. The error bars reflect a 95% confidence interval of the data. The number of Nile rats: *n* = 33 (green), *n* = 172 (red).

**Figure 4 nutrients-13-00094-f004:**
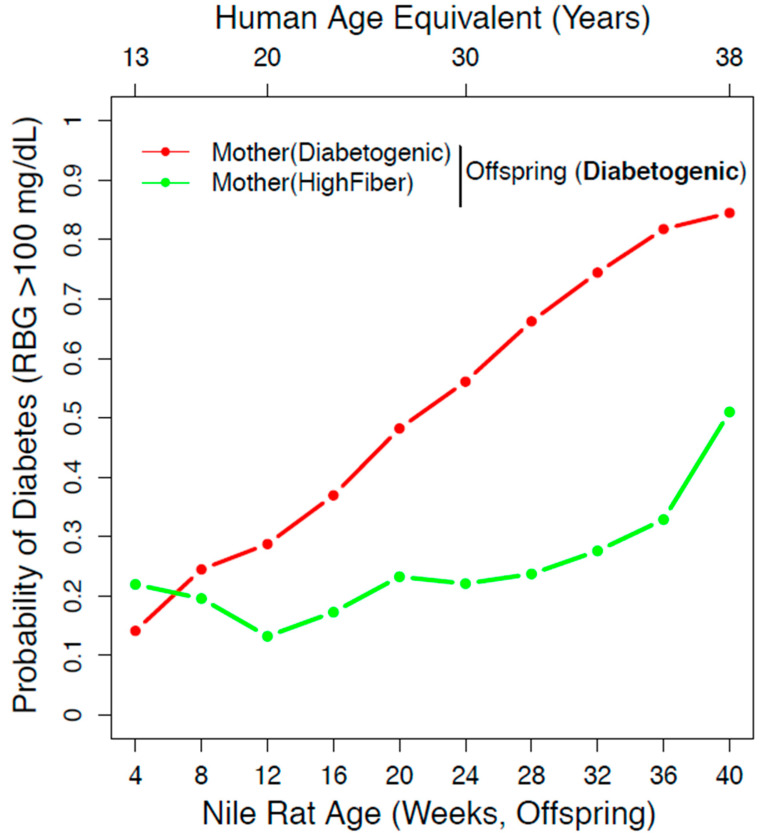
Offspring have a reduced probability of diabetes when mothers consumed a high-fiber diet. We used a simple Bayesian network that incorporated maternal diet, offspring diet, and sex to calculate the probability of diabetes due to maternal diet differences. When the offspring is on a diabetogenic diet, the probability of diabetes is greatly reduced in the group with a maternal high-fiber diet compared to a maternal diabetogenic diet. The number of Nile rats: *n* = 95 (green), *n* = 355 (red).

**Figure 5 nutrients-13-00094-f005:**
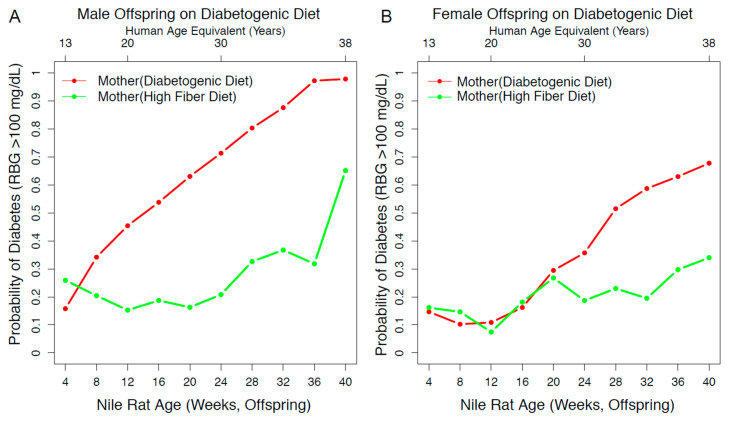
The probability of diabetes due to maternal diet was reduced to a larger extent in the male offspring. (**A**) In male offspring on a diabetogenic diet, the ones from mothers on a high-fiber diet experienced a reduced probability of diabetes from 8 to 40 weeks old. The number of Nile rats: *n* = 62 (green), *n* = 183 (red). (**B**) In female offspring on a diabetogenic diet, the ones from mothers on a high-fiber diet experienced a reduced probability from 25 to 40 weeks old. The number of Nile rats: *n* = 33 (green), *n* = 172 (red).

## Data Availability

The data that support the findings of this study are openly available in Supplementary Data.

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
