# Peer review of "Maternal High-Fiber Diet Protects Offspring against Type 2 Diabetes"

_nutrients, 2020, doi:10.3390/nu13010094_

Round 1
Reviewer 1 Report
This study by Toh et al addresses the matter whether mother´s diet during pregnancy impacts their offspring´s their offspring´s risk of developing diabetes. Once, maternal and/or children malnutrition was proposed to be related to diabetes later in life, mainly observed in some low income countries, e.g. India. However, no obvious link was ever shown to mother´s or children´s diet and today malnutritional diabetes has been omitted as a dignosis by the WHO and the main diabetes associations.
Today, for type 2 diabetes, a diet causing obesity is rather related to diabetes risk.
Toh et al has used the Nile Grass rat as a model of diet-induced diabetes. When in freedom, this rat has a diet based mainly on green plants, is physically active and is not likely to develop any glucose intolerance. When put into a cage with no physical activity and free access to standard lab animal chow, the rats will develop diabetes. Thus, during these circumstances the standard chow is diabetogenic.
Exposure to maternal high fibre diet prior to weaning reduced the risk of developing diet-induced diabetes in the offspring when adult. In addition, offspring pups eating high fibre food after weaning had decreased diabetes risk later in life.
The authors conclude that their investigation demonstrates, in the diabetes-prone Nile rat, that a protective effect of high-fibre-diet can be transmitted to the offspring via maternal food intake. Thus, the authors propose that these findings can have important implications in diabetes prevention.
General comments: The study protocol and methods used seem adequate. The results are interesting and appears to be well interpreted. Against this background it would be even more interesting to investigate if the results in Nile rats are also applicable in humans.
Suggestions:
To Discussion: In this context, and in the absence of a randomised clinical trial it would be pertinent to refer to an observational study on whole grain intake and diabetes risk, e.g. Wirstrom T et al: Consumption of whole grain reduces risk of deteriorating glucose tolerance including prediabetes. Am J Clin Nutr 2013; 97:179.
To Methods: Please add the fibre content of the used lab chows. By looking in the home page of the diet manufacturer, it is stated that the standard chow (Newco Speciality Formulab Diet 5008) contains 3.8% crude fibres. However, it was not possible to find corresponding percentage of the high fibre diet (Lab Diet 5L3M).
It would also be valuable to calculate the total fibre intake per time unit (day). If available, please add the daily intake of food and the calculated fibre intake. Was there a difference in daily food intake related to the type of chow, i.e. standard of high-fibre? This could be relevant for the development of overweight and diabetes.
Reviewer 2 Report
Authors performed a longitudinal study of 671 Nile rats to investigate how early life environment affects the offspring’s diabetic susceptibility through the maternal diet. In this study, they showed that a maternal high fiber diet reduces the probability of diabetes in offspring that consumed a diabetogenic diet after weaning. In addition, offspring consuming a high fiber diet after weaning do not induce diet-induced diabetes, even when exposed to a diabetogenic maternal diet. These results are interesting in essence. There are, however, some points which should be addressed for publication in this journal.
Specific comments
Comment 1: Authors should show the data of food intake for a diabetogenic diet and a high fiber diet in these rats.
Comment 2: Additionally, the data of the changes in body weight should be presented as shown in Figs. 1 and 2.
Comment 3: How about the differences in insulin resistance or insulin secretion between the diabetogenic diet offspring from mothers with a high fiber diet and that from mothers with a diabetogenic diet?
Criticism 4: Is there a difference in maternal body weight or blood glucose levels due to the difference in maternal diet?
Reviewer 3 Report
The manuscript by Toh et al. shows that high fiber diet has a protective effect on diabetogenic diet-induced blood glucose regardless of the consumption start time. The study is interesting and may be important, however the provided evidence is not fully supporting the conclusion and clearly overstated and should be modified.
Specific comments
Composition of diets used should be provided. The duration of mothers on high fiber diet is not clear.
What about the high fiber diet on body weight especially on birth weight. It has any effect on development including fertility.
Food intake or calorie intake should be presented. It is ideal to compare the phenotype after matching calorie intake.
Random blood glucose was the only data presented in this manuscript, so the authors referred this phenotype to type 2 diabetes is not correct. As the authors mentioned in the Introduction, the most important hallmark of type 2 diabetes is insulin resistance. Therefore, the blood insulin level especially glucose-induced insulin level should be measured and glucose tolerance test and insulin tolerance test should be performed to see whether the mice are diabetic. Even the authors claim that random blood glucose is a biomarker for type 2 diabetes but it is better to measure fasting blood glucose.
The beneficial effect of dietary fiber consumption on prevention of insulin resistance and diabetes is not novel, but the mechanisms are not completely understood. Providing possible mechanisms for the observed phenotype would greatly improve the manuscript.
Statistics and number of mice should be indicated in figure legends.
Reviewer 4 Report
In the paper entitled Maternal high fiber diet protects offspring against type 2 diabetes Toh and co workers analyzed the high fiber diet capability in protecting Nile rats offspring from diabetogenic diet effects. Authors provided observational evidence that protective effect of high fiber diet can be transmitted to the offspring and therefore this kind of diet can be considered as an essential prerequisite to avoid type 2 diabetes onset.
This paper is well written, rationale is clear and data are comprehensible and exhaustive. Figure legends are effective and succinct.
This study is interesting and well performed, although some concerns are to be assessed to the Authors:
- As mentioned above, this study has a strong observational nature since the authors do not provide any data to identify a possible molecular mechanism to justify their experimental observation. The scientific soundness of the paper could be dramatically improved, if the Authors would be able to provide this kind of information.
- Nile rats are a frequently used rodent type 2 diabetes model. Could the Authors hypothesize the same behaviour for the offspring of the C57BL6 mice in the same experimental conditions?
- In Materials and Methods section, row 92, Authors better turn housing temperature in Celsius degree instead of Fahrenheit.
- Some mistyping are present in Article Title: authors must remove the “Firstname Lastname” repeated three times (rows 3,4); row 6: delete 1 space between superscript "1" and the word "Neuroscience".
Round 2
Reviewer 2 Report
The authors appropriately responded to my comments in the revision.
Author Response
Thank you!
Reviewer 3 Report
The authors should provide fundamental data including food intake, body weight and blood insulin levels at different time during the study.
